# Does Risser Casting for Adolescent Idiopathic Scoliosis Still Have a Role in the Treatment of Curves Larger Than 40°? A Case Control Study with Bracing

**DOI:** 10.3390/children9050760

**Published:** 2022-05-22

**Authors:** Giovanni Andrea La Maida, Enrico Gallazzi, Donata Rita Peroni, Alfonso Liccardi, Andrea Della Valle, Marcello Ferraro, Davide Cecconi, Bernardo Misaggi

**Affiliations:** U.O. Patologia Vertebrale e Scoliosi, ASST Gaetano Pini-CTO, 20122 Milano, Italy; lamaida.ga@gmail.com (G.A.L.M.); donatarita.peroni@asst-pini-cto.it (D.R.P.); alfonsoliccardi91@gmail.com (A.L.); andrea.dellavalle@asst-pini-cto.it (A.D.V.); marcello.ferraro@asst-pini-cto.it (M.F.); davide.cecconi@asst-pini-cto.it (D.C.); bernardo.misaggi@asst-pini-cto.it (B.M.)

**Keywords:** adolescent idiopathic scoliosis, casting, bracing, conservative treatment

## Abstract

**Background**: The most common conservative treatment for Adolescent Idiopathic Scoliosis (AIS) is bracing. However, several papers questioned the effectiveness of bracing for curves between 40° and 50° Cobb: the effectiveness in preventing curve progression could be as low as 35%. Seriate casting is considered a standard approach in early onset scoliosis; however, in the setting of AIS, cast treatment is seldom utilized, with only few studies reporting on its effectiveness. **Aim of the study:** The main aim of the study is to determine whether a seriate casting with Risser casts associated with bracing is more effective in preventing curve progression than bracing alone in curves larger than 40°. Furthermore, the secondary endpoints were: (1) is there a difference in effectiveness of casting between Thoracic (T) and Thoracolumbar/Lumbar (TL/L) curves? (2) Does the ‘in cast’ correction predicts the treatment outcome? (3) What is the effect on thoracic kyphosis of casting? **Methods**: This is a retrospective monocentric case–control study; through an Institutional Database search we identified all the patients treated at our institution between 1 January 2017 and 31 December 2020, with a diagnosis of AIS, Risser grade between 0 and 4 at the beginning of the treatment, at least one curve above 40° Cobb and treatment with either seriate Risser casting and bracing (Study Group, SG) or bracing alone (Control Group, CG). Standing full spine X-rays in AP and LL are obtained before and after the cast treatment; only AP standing full spine X-rays ‘in-cast’ are obtained for each cast made. Patients were stratified according to the curve behavior at the end of treatment (Risser 5): progression was defined as ≥6° increase in the curve magnitude or fusion needed; stabilization is defined as a change in curve by ±5°; and improvement was defined as ≥6° reduction in the curve. **Results**: For the final analysis, 55 compliant patients (12 M, 43 F, mean age 13.5 ± 1.6) were included in the SG and 27 (4 M, 23 F, mean age 13.6 ± 1.6) in the CG. Eight (14.5%) patients in the SG failed the conservative treatment while 14 (51.3%) failed in the CG. Consequently, the Relative Risk for progression in the Efficacy Analysis was 1.8 (95% CI 1, 3–2.6, *p* = 0.001), and the Number Needed to Treat was 2,4. No significant difference was found between the T and TL/L curves concerning the ‘progressive’ endpoint (z-score 0.263, *p* = 0.79). The mean percentage of ‘in cast’ curve reduction was 40.1 ± 15.2%; no significant correlation was found between the percentage of correction and the outcome (Spearman Correlation Coefficient 0.18). Finally, no significant differences between baseline and end of FU TK were found (32° ± 16.2 vs. 29.6 ± 15.8, *p* = ns). **Discussion:** Seriate Risser casting for AIS with larger curves (>40° Cobb) is effective in reducing curve progression when compared with full time bracing alone in treatment compliant patients. The treatment is equally effective in controlling T and TL/L curves; furthermore, a slight but non-significant decrease in TK was observed in patients treated with casting. This type of treatment should be considered for AIS patients who present with large curves to potentially reduce the percentage of surgical cases. **Short Abstract**: The aim of the study is to determine whether seriate Risser casting associated with bracing is more effective in preventing curve progression than bracing alone in curves larger than 40°. This is a retrospective monocentric case–control study; we identified all the patients treated at our institution with a diagnosis of AIS, Risser grade 0–4 at the beginning of the treatment, at least one curve above 40° Cobb (35° if treated with bracing alone) and treatment with either seriate Risser casting and bracing (Study Group, SG) or bracing alone (Control Group, CG). Fifty-five patients (12 M, 43 F, mean age 13.5 ± 1.6) were included in the SG and 30 (5 M, 25 F, mean age 13.9 ± 1.7) in the CG. Eight (14,5%) patients in the SG failed the conservative treatment while fifteen (50%) failed in the CG. Consequently, the Relative Risk for progression in the Efficacy Analysis was 1.8 (95% CI 1.3–2.6, *p* = 0.001), and the Number Needed to Treat was 2,4. Seriate Risser casting for AIS with larger curves (>40°) is effective in reducing curve progression when compared with full time bracing alone. This type of treatment should be considered for AIS patients who present with large curves.

## 1. Introduction

Adolescent idiopathic scoliosis (AIS) is the most common spinal deformity in growing spine, affecting 1–3% of adolescents. The correct treatment for AIS is weighted on two parameters, namely, teh Cobb Angle of the curve and its risk of progression, either during treatment or in adulthood [1]. Data from natural history study on AIS progression in adulthood show that curves below 35° Cobb at skeletal maturity have a low risk of progression, whereas curves greater than 50° tend to progress in adulthood and are associated with several complications [2,3]. Therefore, the aim of the conservative treatment during growth is to avoid curve progression over 50° Cobb: if the curve reaches this amplitude during growth, surgical treatment is almost always proposed to the patient [4].

In this context, the most common conservative treatment for AIS is bracing: the literature gives evidences in support of bracing effectiveness when compared to observation alone [5,6]. Indeed, published guidelines recommend bracing treatment for curves with an amplitude greater than 25° Cobb during growth, while surgery is considered for curves of amplitude greater than 50° [1,7]. However, several papers questioned the effectiveness of bracing for curves larger than 40° Cobb: while there is great heterogeneity among the inclusion criteria and the reported results, the effectiveness in preventing curve progression could be as low as 35%, especially for patients with curves >50°, with most of the paper reporting an effectiveness around 50% [8,9,10,11]. Seriate casting is considered a standard approach in early onset scoliosis, either as a definitive treatment associated with bracing or as ‘delay’ strategy aimed at controlling the deformity progression until an adequate skeletal maturity is reached and surgery is safer; however, in the setting of AIS, cast treatment is nowadays seldom utilized, with only few studies reporting on its effectiveness [12,13]. Nonetheless, seriated casting treatment is indicated for AIS with curves greater than 40° on the basis of expert’s opinion [1].

Given the scarcity of literature on the use of seriate casting in AIS with curves greater than 40°, we designed a case–control study aimed at answering the following clinical question: is seriate casting with Risser casts associated with bracing more effective in preventing curve progression than bracing alone in curves larger than 40°? Furthermore, the secondary endpoints were: (1) Is there a difference in effectiveness of casting between thoracic and thoracolumbar/lumbar curves? (2) Does the ‘in cast’ correction predicts the treatment outcome? (3) What is the effect on thoracic kyphosis of casting?

## 2. Materials and Methods

### 2.1. Study Design and Population

This is a retrospective monocentric case–control study, level of evidence III. The study was conducted in accordance with the Helsinki Declaration, and Informed Consent for Database Inclusion was obtained for all the patients included in the Study. The Institutional Review Board approval for this study was obtained.

Inclusion criteria for this study were based on the SOSORT Guidelines [1] and are as follows: age > 10 and <18 years; diagnosis of adolescent idiopathic scoliosis; Risser grade between 0 and 4 at the beginning of the treatment; at least one curve above 40° Cobb; treatment with either seriate Risser casting and bracing or bracing alone; good compliance to brace treatment; Risser grade 5 at the end of the follow up.

The study was performed in March 2022; through an Institutional Database search, we identified all the patients treated at our institution between 1 January 2017 and 31 December 2020 that matched the inclusion criteria and retrieved all their medical records. Patients with incomplete medical records were contacted and asked to provide the missing documentation. Each patient was asked whether they underwent spinal fusion at other institutions. On the basis of the medical records, patients were divided in two groups: the Study Group (SG), which included patients who underwent seriate Risser casting followed by brace treatment, and the Control Group (CG), which included patients who underwent treatment with only a brace.

### 2.2. Treatment and Radiological Evaluation

Patients evaluated at our Institution follow the same treatment protocol: In case a seriated casting is proposed, they are admitted to the ward, where the casting is performed without anesthesia on a Risser Frame by an experienced nurse under medical supervision. Following an overnight stay, patients are discharged after a full medical evaluation that checks for GI tract complications and ‘in-cast’ comfort. Full instruction on how to take care of the cast and on how to clean the skin are given to the patients and family at discharge. The cast is maintained for 60–75 days, after which the patients are admitted again; the first cast is then removed and the skin checked: if no skin lesions are found, another cast is made following the same protocol; the full treatment cycle consists of three consecutive casts. At the end of the cast treatment, a Lyon brace is prescribed to be worn full time (23 h per day). Standing full spine X-rays in AP and LL at the beginning of treatment (baseline) and at the final Follow-up (FU, after weaning phase, see below) were obtained; only AP standing full spine X-rays ‘in-cast’ are obtained for each cast made, to confirm the effectiveness of the correction.

Patients who do not accept casting after consultation are prescribed a Lyon brace to be worn full time (23 h a day), and followed accordingly at the outpatient clinic.

Patients were routinely evaluated at follow up visits clinically every 6 months to check well-being and brace congruence and radiographically every 12 months. At each follow-up visit, patients and family were asked about treatment compliance, which was recorded as ‘good’ if brace is worn > 80% of the prescribed time, ‘moderate’ if 80–50% of the time, and ‘poor’ if <50% of the time.

Brace weaning at the end of treatment was the same for the two groups: brace was progressively abandoned in a month, by progressively reducing the daily wearing time.

Cobb angle of the main and secondary curve, as well as Thoracic Kyphosis (TK), were measured on the first X-rays and at final X-rays; Cobb Angle of the curves were also measured on the ‘in-cast’ X-rays to evaluate the casting correction. Final X-ray was the last one after weaning period in case of treatment being stopped for reaching skeletal maturity, or the last X-ray before surgery.

### 2.3. Statistical Analysis

Given that the main aim of this paper was to evaluate whether the seriate casting prevent curve progression, we stratified the patients according to the behavior of the curve from the pre-treatment to the final follow up following the SRS indications [14]: progression was defined as a ≥6° increase in the curve magnitude at the end of treatment or fusion needed; stabilization is defined as a change in curve by ±5°; and improvement was defined as a ≥6° reduction in the curve.

Statistical analysis was performed using VassarStats (Website for Statistical Computation; www.vassarstats.net, accessed on 14 March 2022), while Propensity Score was performed using XLSTAT (Addinsoft Inc., New York, NY, USA). Since continuous variables followed a normal distribution, we used the two tailed Student t-test to compare the baseline characteristics of the SG and CG. Given the observational nature of the study, propensity score was calculated to ensure that the outcome was a consequence of treatment and not of any baseline difference between the two groups. Given that inclusion criteria were the same for both groups, the main confounding variables were the age and Cobb angle of the main curve. Matching was made with the one-to-several approach given the difference in the sample size between the two populations. In order to evaluate the effectiveness of seriate casting, we calculated the Relative Risk (RR) for progression of the curve and the Number Needed to Treat (NNT) for either progression or reduction in the curve; the Confidence Interval (CI) was set at 95%. Some patients included in the SG could not complete the full cycle of three castings due to medical reasons; therefore, to not exclude those from the analysis, we performed both an Efficacy Analysis (EA), including only the patients that completed the treatment, and an Intention to Treat (ITT) analysis for the whole cohort, including in the results those who did not complete the treatment.

The difference in effectiveness of the cast between Thoracic and Thoracolumbar/Lumbar curves was analyzed using Z-Statistic test; Spearman Correlation and Student t-test were used to evaluate the association between the ‘in cast’ correction and the treatment outcome; finally, Student t-test was used to compare baseline and end of FU TK. The significance level was set at 0.05 for all tests.

## 3. Results

After the Database search, we identified 75 eligible patients for the SG and 35 for the CG. However, 20 patients in the SG and 8 in the CG were excluded due to incomplete medical records and/or the inability to contact them; therefore, for the final analysis, 55 patients (12 M, 43 F, mean age 13.5 ± 1.6) were included in the SG and 27 (4 M, 23 F, mean age 13.6 ± 1.6) in the CG (see Figure 1).

Baseline Risser grade, baseline and final FU Cobb Angles for the main and secondary curves, as well as the ‘in cast’ correction, and TK are reported in Table 1. Of the patients included in the SG, 29 (52.8%) had a Risser grade 0–2, and 26 (47.2%) had a Risser grade 3 or 4. In the CG, 15 (55.6%) had a Risser grade of 0–2, and 12 (44.4%) had a Risser grade of 3 or 4. After propensity score matching, all patients were included in the analysis. The Standardized Mean Difference for both the variables analyzed (Age, Main Curve Cobb Angle at Baseline) was below 0.1 before matching.

Concerning the main endpoint of the study, 8 (14.5%) patients in the SG failed the conservative treatment either for progression of the curve (4) or fusion (4, mean Cobb Angle before fusion 51.9 ± 4.7°), while 14 (51.3%) failed in the CG (10 progressed, 4 fused at a mean Cobb Angle before fusion of 50.2 ± 3.6°); in the SG, 31 (56.4%) patients were nonprogressive, and 16 (29.1%) improved (Figure 2), while in the CG, the nonprogressive and improved patients were 12 (44.4%) and 1 (3.7%), respectively. Eight patients (14.5%) in the SG were unable to complete the full treatment cycle due to skin lesions. Consequently, the RR for progression in the EA was 1.8 (95% CI 1.3–2.6, *p* = 0.001), and the NNT was 2.4. In the ITT analysis, the RR for progression was 1.7 (95% CI 1.2–2.5, *p* = 0.005), and the NNT was 2.8. In other words, for every 100 patients with seriate casting, 42 (EA) or 35 (ITT) would not fail when compared to brace only.

Concerning the effectiveness of casting between T and TL/L curves, there were 5 (9%) patients with a TL/L curve only, and 18 (32.7%) patients with a double curve, for a total of 23 TL/L curves and 50 main Thoracic curves. No significant difference was found between the two types of curves concerning the ‘progressive’ endpoint (z-score 0.263, *p* = 0.79). Concerning the ‘in-cast’ correction, the mean percentage of ‘in-cast’ curve reduction was 40.1 ± 15.2%; no significant correlation was found between the percentage of correction and the outcome (Spearman Correlation Coefficent 0.18); nonetheless, the mean percentage of correction in the progressive group was 32.8 ± 12.6%, which is lower, albeit not significantly, than the percentage for the stable (40 ± 15.3%, *p* = ns) and the improved groups (0.42 ± 15.5, *p* vs. failed = ns). Finally, no significant differences between baseline and end of FU TK were found (32° ± 16.2 vs. 29.6 ± 15.8, *p* = ns).

## 4. Discussion

The present study was designed to evaluate the effectiveness of seriate Risser casting in AIS with large curves. Our main finding is that compliant patients treated with a full-time brace only have a Relative Risk for curve progression of 1.8 (Efficacy Analysis) when compared with seriate casting followed by full-time brace, thus proving the effectiveness of this approach. In other words, for every 100 patients treated with seriate casting, there would be 42 less curve progressions when compared with standard treatment with brace only. Interestingly, the effectiveness of the brace-only treatment in our study is comparable to what was previously reported in similar cohorts [11,15]. Furthermore, we showed how casting is equally effective in preventing progression of Thoracic and Thoracolumbar/Lumbar curves, thus being applicable regardless of curve location.

This study has several limitations. First and foremost, the retrospective nature of this analysis is a design limitation; however, all patients treated in a definite timespan were included in the initial selection, and those who were excluded due to incomplete medical records are a minority. Furthermore, as already described in other prospective studies, patients treated with long-term treatment such as braces and casts often refuse further follow up due to psychological reasons; thus, a certain amount of incomplete follow up is expected [9]. Second, treatment compliance was evaluated by relying on self-reported compliance by patients and families. Treatment compliance is one the factors that influences the effectiveness of conservative treatment the most [16,17]. Therefore, we only included patients with self-reported good compliance in the study. While we are aware that this method could overestimate the number of good compliers, other papers with large cohorts found that self-reported good compliance is still a good predictor of treatment effectiveness [16]. Indeed, in our study, the effectiveness of brace treatment in CG is comparable to what was previously reported [9,15]. Finally, no formal evaluation of psychological impact of either bracing and casting was performed. The psychological impact of bracing and casting were extensively evaluated in literature, with a reported negative impact on quality of life and treatment satisfaction in more than 50% of patients treated with full-time bracing [18,19]. This is especially true for casting, which cannot be removed for the duration of the treatment and thus limits the access to sport and other recreational activities [18]. Nonetheless, no patient included in this study asked for the casting treatment to be stopped for psychological reasons. In our department, patients and families receive a full consultation before being enrolled in casting, with a detailed explanation on the necessity of this treatment and of the limitation imposed. This highlights the importance of counselling and family support to achieve good results with this treatment.

The effectiveness of bracing in AIS is well-documented in the literature: several papers, including randomized controlled trials, have shown how bracing can prevent curve progression [5,20,21]. However, most of the studies focus on a specific subset of the population, i.e., patients with curves between 20° and 40° and a Risser grades of 0 to 2 [22]. On the other hand, for patients with curves larger than 50°, surgery is always indicated, since bracing alone is considered ineffective in correcting and reducing scoliosis curves [7]. Therefore, patients with Cobb angles between 40° and 50° constitute a ‘grey zone’, where no treatment has proven to be completely effective: on one side, conservative treatment with bracing alone has shown conflicting results [8,9,11,15]; on the other, even surgery has been questioned, with reported medium-term results comparable to those of conservative treatment [23]. Seriate casting is a staple of Early Onset Scoliosis treatment, and in most of Western countries, it is nowadays used only in this specific setting. However, casting in AIS was widely used in the past, although its effectiveness is poorly reported in the recent literature [12,24]. Historically, conservative treatment for AIS consisted in an ‘active’ correction phase, followed by a ‘stabilization’ phase [25]. When compared to bracing, casting has the possibility of modulating the corrective forces acting in conjunction on spinal elongation through traction on Risser frame and lateral push; this allows for an ‘active’ correction of the curve, as we have shown with a mean percentage of ‘in cast’ curve correction of 40%. In the present study, we have indeed shown how an ‘active’ correction of the curve followed a ‘stabilization’ through the use of brace is significantly more effective than stabilization with bracing alone in preventing curve progression.

One of the main reasons for progressive abandonment of casting was the high impact on patient’s quality of life; among the main limiting factors, the occurrence of skin lesions was one of the most important [18]. In our study, only 14.5% of patients developed skin lesions severe enough to stop the treatment. However, even including those patients in an ITT analysis, the effectiveness of cast treatment was still superior to bracing alone in preventing curve progression.

Two other concerns on conservative treatment are its effectiveness on TL/L curves and its perceived flattening effect on TK. When correcting a T curve with a cast (or brace), the corrective force is applied though the ribs and thus reaches the thoracic spine through a ‘rigid’ structure, with a theoretical more effective force transfer on the spine than in lumbar curves. However, we found no significant difference in progression prevention between T and TL/L curves, with similar correction percentages. Our results are consistent with other studies on bracing, that report a similar correction between T and TL/L curves [24,26]. With casting, this observation could be explained by the corrective effect of the traction on Risser frame, which is greater on the more mobile lumbar spine. Finally, concerning the effect on the sagittal profile, several studies on bracing reported a flattening effect on the TK during the treatment [27,28]. This observed effect has been explained as a consequence of the forward push applied on the thoracic rib hump [29]. In our study we found a slight but non-significant decrease in TK between the baseline X-rays and the final FU; on the contrary, other papers reported a significant reduction in TK with conservative treatment when comparing the baseline with the reaching of skeletal maturity [30]; our results could depend on the decision to measure the overall TK, which other studies have reported as unchanged, rather than the T4-T12 TK [26], or by the insufficient sample size of our paper to detect the magnitude of change, which is reported in the range of 3 to 4° Cobb [30,31].

## 5. Conclusions

The present study is one of the few reports on the effectiveness of seriate casting in preventing AIS curve progression, and the only one from the last 15 years. We have shown how seriate Risser casting for AIS with larger curves (>40° Cobb) is effective in reducing curve progression when compared with full-time bracing alone in treatment-compliant patients. While we recognize that this type of treatment is demanding for patients and family, we observed only a small percentage of patients that were unable to complete the treatment. The treatment is equally effective in controlling T and TL/L curves; a slight but non-significant decrease in TK was observed in patients treated with casting. This type of treatment should be considered for AIS patients who present with large curves to potentially reduce the percentage of surgical cases.

## Figures and Tables

**Figure 1 children-09-00760-f001:**
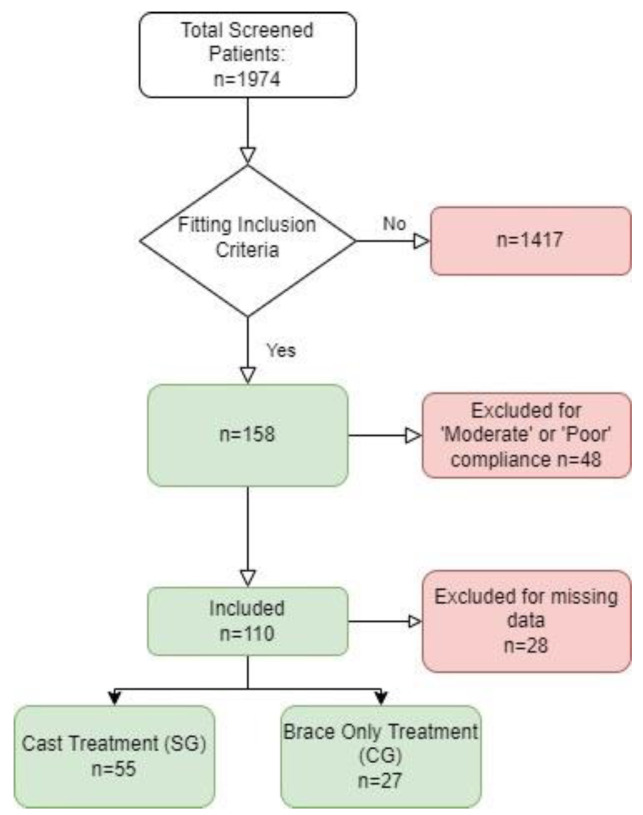
Flow chart for Patient’s inclusion in the present study.

**Figure 2 children-09-00760-f002:**
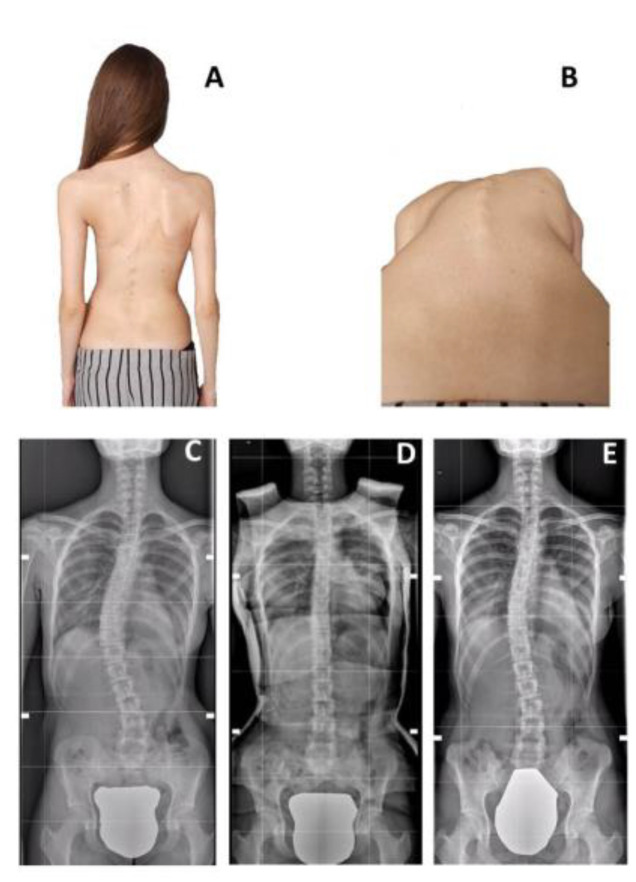
Case Example for an improved patient in the SG (12 YO Female). Panel (**A**,**B**) shows the clinical picture before casting; rib hump is clearly visible; Panel (**C**) shows the baseline X-ray, with a 41° T Curve, R = 0; Panel (**D**) shows the in-cast correction; Panel (**E**) shows the final FU X-ray, with the curve reduced to 26°, R = 5.

**Table 1 children-09-00760-t001:** Baseline, in-cast, and post-treatment characteristics for the two groups.

	Study Group	Control Group	*p*
*Age*	13.5 ± 1.6	13.5 ± 1.6	ns
*Risser*	1.4 ± 1.6	2.0 ± 1.6	ns
*Treatment duration (year)*	3.4 ± 0.5	3.8 ± 1.1	ns
*Main Curve (baseline)*	42.7 ± 9.5 (40.1–65.1) ^Δ^	43.6 ± 7.8 (40.5–66.7) ^ø^	ns
*Main curve (in-cast)*	27.5 ± 9.1		
*Main Curve (Final FU)*	39.7 ± 11.7 ^Δ^	45.1 ± 10.5 ^ø^	ns
*Secondary Curve (baseline)* *	41.4 ± 6.8 (30.1–55.1) ^θ^	40.6 ± 6.5 (24.1–47.5) ^β^	ns
*Secondary curve (in-cast)*	28.7 ± 10.5		
*Secondary Curve (Final FU)*	41.8 ± 8.2 ^θ^	42.5 ± 8.1 ^β^	ns
*TK (baseline)*	32 ± 16.2	29.7 ± 14.4	ns
*TK (Final FU)*	29.6 ± 15.8	27.2 ± 15.8	ns

* 18 patients in the SG and 9 in the CG had a double curve. Values are reported as Means ± SD (range). ^Δ^, *t*-test *p* = ns; ^θ^, *t*-test *p* = ns; ^ø^, *t*-test *p* = ns; ^β^, *t*-test *p* = ns.

## Data Availability

Raw data for this study are available on request to the Corresponding Author.

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
