# Peer review of "Does Risser Casting for Adolescent Idiopathic Scoliosis Still Have a Role in the Treatment of Curves Larger Than 40°? A Case Control Study with Bracing"

_children, 2022, doi:10.3390/children9050760_

Round 1

Reviewer 1 Report

Dear Author.

I read your manuscript "

oes Risser Casting for Adolescent Idiopathic Scoliosis still have a role in the treatment of curves larger than 40°? A case control study with bracing " with great attention and I think that some clarifications and improvements could be added: 1. Inclusion criteria Risser 0-4. - That is beyond any acceptable standard, please provide explanation to the reasoning for bracing /casting(treatment initiation) for patients with Risser beyond 2. 2. Selection Bias- Potential significant room for bias as it is not described how patients were directed to those groups, what is the institutional policy for conservative treatment and how the method is decided, is it a parent decision or a providers decision, any chance for clear criteria about that decision? 3. 20 patients were omitted, I would like their inclusion dat be presented beyond the fact that their records couldn't be achieved, that would demonstrate once again lack of bias. 4. The author comments about HRQOL that is a concern, is it measured or addressed in any way? do you have some data about that? 5. When exactly x rays are done? in other words, when the author decided if treatment is successful of failed. Was it immediately after removal of brace/cast? or there was a minimum follow up time ?

Author Response

Dear Reviewer 1,

First of all, thank you very much for your comments and for the opportunity to improve our paper.

Does Risser Casting for Adolescent Idiopathic Scoliosis still have a role in the treatment of curves larger than 40°? A case control study with bracing " with great attention and I think that some clarifications and improvements could be added:

  1. Inclusion criteria Risser 0-4. - That is beyond any acceptable standard, please provide explanation to the reasoning for bracing /casting(treatment initiation) for patients with Risser beyond 2.

Thank for your comment. This indeed highlight an ongoing debate in the scientific community. We recognize that SRS indication for Bracing in Adolescent Idiopathic Scoliosis, based mostly on the inclusion criteria used in the well-known trial by Dolan et al (1), recommend bracing initiation only when Risser is 0-2 (2). On the other hand, the most recent SOSORT guidelines (3) recommend bracing initiation (and even casting) up until Risser 4, especially with larger curves, hence the inclusion criteria for this paper. Furthermore, most European based literature on conservative treatment follows the same criteria as we used in the present paper. We better detailed this in the Methods.

  1. Selection Bias- Potential significant room for bias as it is not described how patients were directed to those groups, what is the institutional policy for conservative treatment and how the method is decided, is it a parent decision or a providers decision, any chance for clear criteria about that decision?

Thank you very much for your comment. Patients fitting th inclusion criteria for this paper are offered both treatment strategies. After an in-depth consultation with the physician, evaluating benefits and limitation of both strategies, a decision is made. Further consultation are done at every follow up visit. We better clarified this in the Methods.

  1. 20 patients were omitted, I would like their inclusion dat be presented beyond the fact that their records couldn't be achieved, that would demonstrate once again lack of bias.

Thank for your comment. Data for those patients are largely incomplete (baseline X-Rays not in our PACS to confirm measurement, missing FU visits, treatment compliance data not available), therefore we do not think that presentation of those data is worth. Nonetheless, we performed an additional statistics (Propensity Score Matching) to further reduce potential bias of a retrospective observational study.

  1. The author comments about HRQOL that is a concern, is it measured or addressed in any way? do you have some data about that?

Thank you for your comment. No ‘formal’ evaluation of HRQOL and psychological impact of treatment was conducted in this study. Nonetheless, we recognize this as a limitation, and added this in the discussion.

  1. When exactly x rays are done? in other words, when the author decided if treatment is successful of failed. Was it immediately after removal of brace/cast? or there was a minimum follow up time ?

Thank you for your comment. We detailed in the Methods that success or failure are defined at the end of treatment, i.e. at the last X-Ray performed when conservative treatment is stopped (either for skeletal maturity or for progression to Surgery) (see Results, page 4, line 178-180). Nonetheless, we better clarified this also in the Methods.  

References

  1. Dolan LA, Wright JG, Weinstein SL. Effects of bracing in adolescents with idiopathic scoliosis. Vol. 370, The New England journal of medicine. United States; 2014. p. 681.
  2. Roye BD, Simhon ME, Matsumoto H, Bakarania P, Berdishevsky H, Dolan LA, et al. Establishing consensus on the best practice guidelines for the use of bracing in adolescent idiopathic scoliosis. Spine Deform. 2020 Aug;8(4):597–604.
  3. Negrini S, Donzelli S, Aulisa AG, Czaprowski D, Schreiber S, de Mauroy JC, et al. 2016 SOSORT guidelines: orthopaedic and rehabilitation treatment of idiopathic scoliosis during growth. Scoliosis spinal Disord. 2018;13:3.

Reviewer 2 Report

This is a retrospective review of a one-institution population on an interesting topic concerning conservative treatment of relatively large scoliotic curves. The problem of selection bias is present but seems well addressed. I assume, but this could be specified, that the cast is applied without anaesthesia. There are no standards for the amount of correction that can be obtained either in a brace or in a cast. The in-cast correction was not very high at average 40%, the in-brace correction is not mentioned. It is suggested in the text (line 227-229) that bracing is meant more for curve stabilization only whereas casting, followed by bracing, is meant to provide already immediate correction. Maybe the authors can elaborate on their technique, indicating what is considered satisfactory in-brace correction. 

Author Response

Dear Reviewer 2,

First of all, thank you very much for your comments and for the opportunity to improve our paper.

This is a retrospective review of a one-institution population on an interesting topic concerning conservative treatment of relatively large scoliotic curves. The problem of selection bias is present but seems well addressed.

Thank you very much for your comment. To further reduce the issue of selection bias, we added the Propensity Score Matching analysis to our Methods and Results

I assume, but this could be specified, that the cast is applied without anaesthesia.

Thank you very much. Yes, the cast were performed without anaesthesia. We specified in the Methods

There are no standards for the amount of correction that can be obtained either in a brace or in a cast. The in-cast correction was not very high at average 40%, the in-brace correction is not mentioned. It is suggested in the text (line 227-229) that bracing is meant more for curve stabilization only whereas casting, followed by bracing, is meant to provide already immediate correction. Maybe the authors can elaborate on their technique, indicating what is considered satisfactory in-brace correction.

Thank you for your comment. As discussed, the aim of bracing is not to obtain curve correction, but rather to prevent curve progression. Therefore, we do not perform ‘in-brace’ X-Rays, and rely on ‘out-brace’ X-Rays for control of curve progression. The results of our brace treatment for the curves included in this study (>40° Cobb) are in line with those reported in other papers. We clarified better this point in the Discussion

Reviewer 3 Report

The authors have carried out an excellent study, of clinical and scientific interest. The paper is written in a clear, brief, concrete and easy to understand way. The methods and results are clearly shown, as are the conclusions. My congratulations to the researchers for their useful study.

Author Response

Dear Reviewer 3,

Thank you very much for your appreciation of our paper. 

Round 2

Reviewer 1 Report

No further comments

Author Response

Thank you very much for your work and for the oportunity to improve our paper